# Decrease in radiative forcing by organic aerosol nucleation, climate, and land use change

Jialei Zhu [1], Joyce E. Penner[1], Fangqun Yu[2], Sanford Sillman[1], Meinrat O. Andreae [3,4,5] & Hugh Coe[6]

Organic nucleation is an important source of atmospheric aerosol number concentration, especially in pristine continental regions and during the preindustrial period. Here, we improve on previous simulations that overestimate boundary layer nucleation in the tropics and add changes to climate and land use to evaluate climate forcing. Our model includes both pure organic nucleation and heteromolecular nucleation of sulfuric acid and organics and reproduces the profile of aerosol number concentration measured in the Amazon. Organic nucleation decreases the sum of the total aerosol direct and indirect radiative forcing by 12.5%. The addition of climate and land use change decreases the direct radiative forcing ($-0.38 \, \mathrm{W \, m^{-2}}$) by 6.3% and the indirect radiative forcing ($-1.68 \, \mathrm{W \, m^{-2}}$) by 3.5% due to the size distribution and number concentration change of secondary organic aerosol and sulfate. Overall, the total radiative forcing associated with anthropogenic aerosols is decreased by 16%.

[1] Department of Climate and Space Sciences and Engineering, University of Michigan, Ann Arbor, MI 48109, USA. [2] Atmospheric Sciences Research Center, State University of New York at Albany, Albany, NY 12203, USA. [3] Biogeochemistry Department, Max Planck Institute for Chemistry, Mainz, Germany. [4] Scripps Institution of Oceanography, University of California San Diego, La Jolla, CA 92093, USA. [5] Department of Geology and Geophysics, King Saud University, Riyadh, Saudi Arabia. [6] School of Earth and Environmental Sciences, University of Manchester, Manchester M13 9PL, UK. Correspondence and requests for materials should be addressed to J.E.P. (email: penner@umich.edu)

A erosols play a key role in determining the Earth's radiation budget, both directly by absorbing and scattering radiation and indirectly by altering the albedo of clouds[1]. The effects of aerosols on climate constitute the largest uncertainty in the current understanding of climate change[2]. Climate and land use change contribute to this uncertainty by causing changes in biogenic volatile organic carbon (BVOC) emissions that form secondary organic aerosol (SOA) particles. SOA often represents a major fraction of submicron-sized atmospheric organic aerosol and dominates ambient aerosol in tropical and remote regions[3,4].

Nucleation of atmospheric vapors is the largest source of atmospheric aerosol number concentration[5], and is thought to contribute to up to half of the global cloud condensation nuclei (CCN)[6]. Most models only use a particle formation rate based on the concentration of sulfuric acid[7,8]. However, these models fail to reproduce nucleation rates in ambient observations with a low concentration of sulfuric acid[7,9]. Recently, highly oxygenated molecules (HOMs) with extremely low volatility were shown by experiments in the CLOUD (Cosmics Leaving Outdoor Droplets) project to be able to form new particles even in the absence of sulfuric acid[10] and to drive initial particle growth[11]. Ion-induced nucleation of pure organic particles is expected to be a potentially widespread source of new aerosol particles in pristine regions as well as in the preindustrial (PI) atmosphere[10,12].

The aerosol number concentration and size distribution are important factors determining the CCN concentration and largely determine the aerosol indirect effect (AIE)[13]. A small AIE ($<-0.1\,\mathrm{W\,m^{-2}}$ for the global average) of SOA was estimated without organic nucleation involving either pure organics or a heteromolecular process involving organics and sulfuric acid, since most SOA is internally mixed with particles in the accumulation mode that are already CCN[14,15]. A much stronger AIE of SOA ($-0.22$ to $-0.77\,\mathrm{W\,m^{-2}}$) associated with including new particle formation from BVOC and sulfuric acid was calculated due to the increase in the number of CCN-size particles[9,16]. Gordon et al.[12] indicated that pure organic nucleation enhances the baseline of particle number concentrations in the PI environment, resulting in a 27% decrease in anthropogenic aerosol radiative forcing (RF)[12]. However, previous model studies applied empirical or semi-empirical fixed and instantaneous HOM yields[12,16,17]. These studies assumed that autoxidation occurs rapidly and produces HOMs after the initial oxidation step based on the CLOUD project experiments. The latter took place under specific chamber conditions that may not exactly reflect what occurs in the atmosphere. This assumption affects the vertical and horizontal distribution of newly formed SOA as well as the time scales for SOA formation, and it places most new particles near the surface, where BVOC concentrations are highest. In addition, most studies that estimated RF omitted the influence of climate and land use change on the emissions of BVOC, so they are unable to properly represent the change of SOA between the present-day (PD) and PI atmosphere and thus are unable to estimate RF accurately.

This study uses a comprehensive nucleation scheme that includes the heteromolecular, pure neutral, and ion-induced organic nucleation of HOMs that are produced through an explicit chemical mechanism[18–21] instead of the assumption of fast autoxidation in previous studies. Using this model, we are able to reproduce observations of the vertical distribution of condensation nuclei in the Amazon. The influence of organic nucleation as well as climate and land use change on aerosol RF is calculated. Overall, we find a 16% reduced radiative forcing associated with anthropogenic aerosols with the combined effect of organic nucleation, climate, and land use change.

## Results

**Organic nucleation and aerosol number.** The model-calculated global average burden of SOA in the PD is $1.90\,\mathrm{mg\,m^{-2}}$ (Supplementary Fig. 1). The organic carbon (OC) and aerosol number concentration are evaluated in Supplementary S1. BVOC emissions are largest in the tropics, causing the SOA in models to peak there (Supplementary Fig. 1)[17,22]. However, there are only a few measurements available for model validation in this region[17]. New particles formed from BVOC can vastly change aerosol number concentrations, which is one of the most important factors determining aerosol radiative effects. Gordon et al.[12] overestimated CCN concentrations by about a factor of two at the surface in the Amazon. In previous studies, organic nucleation mainly occurs within approximately 500 m of the surface with little change to particle formation above the planetary boundary layer (PBL)[9,12]. However, nucleation within the Amazon PBL is almost never observed[3,23–25]. Moreover, airborne observations over the Amazon Basin show that there are high aerosol number concentrations in the upper troposphere (UT) between 8 and 15 km altitude with concentrations often exceeding those in the PBL by 1 or 2 orders of magnitude[26]. Our explicit chemical mechanism allows BVOC to be transported to the UT, where further oxidation forms extremely low volatility HOMs. The low temperatures, high ionization rates, and a low condensation sink in the UT offer a conducive environment for nucleation. As a result, the profile of aerosol number concentration in the Amazon region (dashed box, Supplementary Fig. 1) peaks near 12 km reproducing these airborne observations (Fig. 1a). The number concentration shown in Fig. 1a is dominated by Aitken mode particles (>95%) formed from the growth of nucleation-mode particles[26]. The predicted $OA/SO_4$ ratio in the UT aerosol over the Amazon is 10.3, comparable to the observed mean value $13.8 \pm 4.6$[26]. As shown in Fig. 1b, more than 80% of the total Aitken particle number is initially formed as a result of organic nucleation with the percentage increasing with altitude, while nucleation of sulfate represents only about 10%.

The model follows the formation of new particles from organics as a result of three mechanisms: heteromolecular nucleation of sulfuric acid and organics (HET), neutral organic nucleation (NON) and ion-induced organic nucleation (ION). The mass of SOA formed as a result of organic nucleation (newSOA) and the organics that condense on them explains 17% of the total global average burden of SOA with a high percentage (>30%) in some oceanic and remote regions (Supplementary Fig. 2). The Amazon is the most important region producing new organic particles and has the highest organic nucleation rate primarily because of the high emissions of α-pinene and the high production of HOMs from α-pinene oxidation[19] (Supplementary Fig. 3a). Over 90% of newSOA over the Amazon is associated with ION, while globally it explains 60.0% (Supplementary Fig. 3d). However, HET occurs more widely than ION and dominates the rate of organic nucleation in many areas (Supplementary Fig. 3b). NON is responsible for only 1.1% of newSOA. Organic nucleation increases the global average newSOA number by $9425\,\mathrm{cm^{-3}}$ and the number concentration in the PBL by $6572\,\mathrm{cm^{-3}}$ (Supplementary Table 1). The column number of newSOA in the nucleation mode is highest in the Amazon (Supplementary Fig. 4a) but the highest number concentration in the PBL occurs downwind of the Amazon (Supplementary Fig. 4b). However, fewer newSOA particles grow to the accumulation mode size in the tropics than in most other regions (Supplementary Fig. 4e). The average mass ratio of condensed sulfuric acid to organics in each of the three modes is shown in Supplementary Fig. 5. The global average mass of sulfuric acid in the nucleation and Aitken mode particles in newSOA is larger than that of organics. Thus, condensed sulfuric acid and water are responsible for most of the growth of small newSOA

particles after organics drive the formation of new particles, consistent with some laboratory and field measurements[27,28]. Organics also contribute to the growth of the small particles,

especially over the Amazon with its high concentrations of semi-volatile organic compounds (SVOC), so that the concentration of organics on newSOA is comparable to that of sulfuric acid. As newSOA particles grow, the decrease of the Kelvin effect with increasing size allows relatively more SVOC to condense in regions with large organic sources[11]. This allows the mass of organics in the accumulation mode of newSOA to become larger than that of sulfuric acid except for regions impacted by pollution in the Northern Hemisphere (NH, Supplementary Fig. 5c).

**SOA in the preindustrial atmosphere.** Organic nucleation provides a way to form particles in the PI atmosphere with its low levels of sulfuric acid, thereby changing the baseline for determining the RF by anthropogenic aerosols[12,26]. The uncertainty in aerosol concentrations in the PI atmosphere is the largest contributor to the uncertainty in estimates of RF[29]. Previous studies have determined the aerosol concentration in the PI atmosphere by only considering the change in anthropogenic emissions[12,30]. However, the source of SOA is very sensitive to changes in climate and land use[14]. Here we calculate SOA in the PI atmosphere including changes in anthropogenic emissions, climate, and land use (PIall) and compare our results with simulations that only include changes in anthropogenic emissions (PIemi). We also compare simulations with and without organic nucleation (Table 1). The net change of natural precursor emissions (i.e., isoprene, α-pinene, and limonene) reflects the integrated competition between increased surface temperature and decreased vegetation (Supplementary Fig. 6), while the natural emissions in PIemi are similar to those in the PD. The emissions of α-pinene increase in most areas in the Amazon and central Africa where organic nucleation is most efficient (Supplementary Fig. 6b). Overall, this causes the global vertically averaged pure organic nucleation rate to increase by 44% in the PD compared to PIall (Supplementary Fig. 7). There are many more newSOA particles growing to the Aitken and accumulation modes in the PD than the PI due to the significant increase of sulfuric acid in the PD. As a result, the column and PBL number concentration of SOA in the Aitken mode increase by 19 and 43% in the PD compared to PIall, while the column and PBL number concentration of SOA in the accumulation mode in the PD is larger than that in PIall by factors of 11.4 and 7.6, respectively (Supplementary Fig. 8). The large increase in the number concentration of accumulation mode newSOA in the PD may be caused by the underestimation of SOA in our model (Supplementary S1), which leads to a very low baseline in the number of newSOA in the accumulation mode in

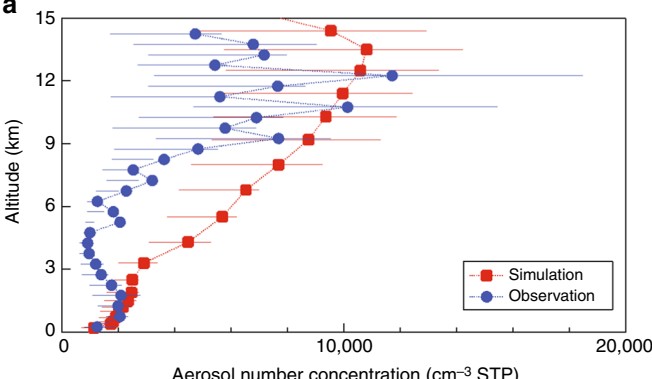

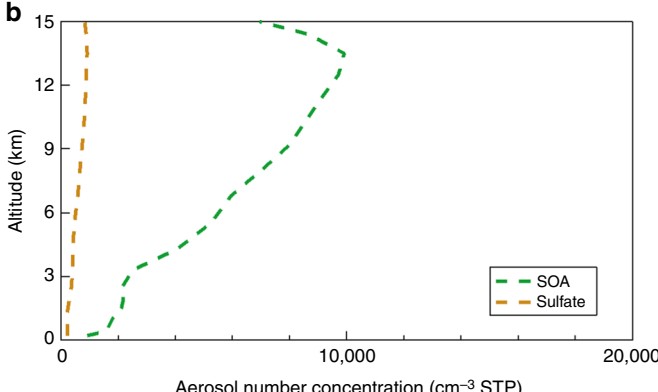

**Fig. 1** Simulated aerosol number concentration and aerosol composition over the Amazon. **a** Comparison of the simulated multi-year average September and October mean total aerosol number concentration over the Amazon and the observed vertical profile from ref. [26]. Both the model results and observations are those with diameters above 10 nm in the lower troposphere and 20 nm in the upper troposphere. The error bar indicates the 25th and 75th percentile of aerosol number concentration for each height layer. **b** The number concentration of newly nucleated sulfuric acid and secondary organic aerosol (newSOA)

**Table 1 Summary of the global average SOA burden, number and radiative effects in PD, PIemi and PIall schemes with and without organic nucleation (ON)**

| | | PD | | PIemi | | PIall | |
|---|---|---|---|---|---|---|---|
| | | w/ ON | w/o ON | w/ ON | w/o ON | w/ ON* | w/o ON |
| Aerosol burden (mg m⁻²) | newSOA (nucleation) | 0.014 | / | 0.031 | / | 0.036 | / |
| | newSOA (Aitken) | 0.16 | / | 0.21 | / | 0.19 | / |
| | newSOA (accumulation) | 0.15 | / | 0.61 | / | 0.81 | / |
| | Total SOA | 1.90 | 2.05 | 1.91 | 2.53 | 2.16 | 2.91 |
| Column aerosol number (10¹⁰ m⁻²) | newSOA (nucleation) | 19438 | / | 32283 | / | 36356 | / |
| | newSOA (Aitken) | 4816 | / | 4103 | / | 4059 | / |
| | newSOA (accumulation) | 27.3 | / | 2.8 | / | 2.4 | / |
| Radiation effect (W m⁻²) | DRE of SOA | −0.148 | −0.118 | −0.171 | −0.238 | −0.203 | −0.278 |
| | AIE of SOA | −0.148 | −0.071 | −0.336 | −0.128 | −0.308 | −0.127 |
| | DRF | / | / | −0.410 | −0.500 | −0.384 | −0.469 |
| | IRF | / | / | −1.735 | −1.952 | −1.675 | −1.878 |

Total SOA includes newSOA and SOA internally mixed with other preexisting aerosols
The slash (/) symbol indicates not available
*ON* organic nucleation, *PD* present day, *w/* with, *w/o* without, *SOA* secondary organic aerosol, *DRE* direct radiative effect, *AIE* aerosol indirect effect, *DRF* direct radiative forcing, *IRF* indirect radiative forcing

PIall. The tropospheric burden of ozone is increased by 33 and 46% in the PD compared to PIemi and PIall, which is similar to the difference between the PD and 1850 from ensemble multi-model results (~30%)[31]. Due to the smaller oxidation rate and smaller concentration of sulfuric acid in PIemi, the HET and ION nucleation rates in PIemi are 78 and 5% smaller than those in the PD, although the emission of α-pinene is similar. The column number concentration of Aitken and accumulation mode newSOA in PIemi are 1.1% and 16.7% larger than those in PIall because of the larger α-pinene emission in the tropics in the PIemi compared to PIall. However, the burden of total SOA in PIall is 14 and 13% larger than that in PD and PIemi, respectively, (Supplementary Fig. 9) due to the larger emission of SOA precursors, especially that of isoprene, in PIall (Supplementary Fig. 6).

**Influence of SOA on radiative forcing.** Here we evaluate the radiative effects of SOA with and without organic nucleation and with and without climate and land use change in both the PD and PI atmosphere. Previous studies did not evaluate the influence of organic nucleation on the direct radiative forcing (DRF)[12]. Our model indicates that the PD direct radiative effect (DRE) of SOA is 25% larger due to organic nucleation (Table 1). The burden of total SOA in PIall is 13% larger than that in PIemi, resulting in the 19% larger global average DRE of SOA in PIall compared to PIemi (Supplementary Fig. 10). The size distribution of newSOA also influences the DRE of SOA. The burden of newSOA contributes 45 and 48% to the total SOA burden in PIemi and PIall, respectively. Although the burden of newSOA in PIall is 22% larger than that in PIemi, the total surface area of newSOA in PIall is only 3.4% larger than that in PIemi due to changes in the

size distribution. Moreover, the difference in organic nucleation caused by climate and land use change can alter the concentration of sulfate indirectly. The sulfuric acid condensed on newSOA (SO4SOA) is decreased by 4% in PIall compared to PIemi (Supplementary Table 2). As a result, the additional sulfuric acid gas in PIall promotes new particle formation of sulfate and the growth of more new sulfate particles to a larger size. The column number concentration of new sulfate in the Aitken and accumulation modes in PIall is 15% and 4.5% larger than that in PIemi, respectively (Supplementary Table 2). As a result, the DRF of anthropogenic aerosol is estimated to be $-0.384\,\mathrm{W\,m^{-2}}$ using PIall but $-0.410\,\mathrm{W\,m^{-2}}$ using PIemi (Fig. 2a, b). Thus, the anthropogenic aerosol DRF is decreased by 6.3% when changes in climate and land use are included. The difference in DRF of anthropogenic aerosols comes from the combined effects of changes in the total SOA burden and the size distributions of new sulfate particles and newSOA. The total surface area of new sulfate and newSOA in PIall is 7.8% larger than that in PIemi, causing a smaller forcing. Without organic nucleation, the DRF of anthropogenic aerosol using PIall is $-0.469\,\mathrm{W\,m^{-2}}$, which is 22% larger than the DRF with organic nucleation (Supplementary Fig. 11). This is because the total new aerosol surface area decreases by 17% in the PI atmosphere due to the decrease in the total aerosol number concentration.

The AIE of SOA is much more sensitive to the inclusion of organic nucleation since it strongly depends on the change in the aerosol number concentration as well as the size distribution. The AIE of SOA is small without organic nucleation (Table 1). Organic nucleation produces newSOA and some grow to CCN size and increase cloud droplet number concentrations (CDNC). In

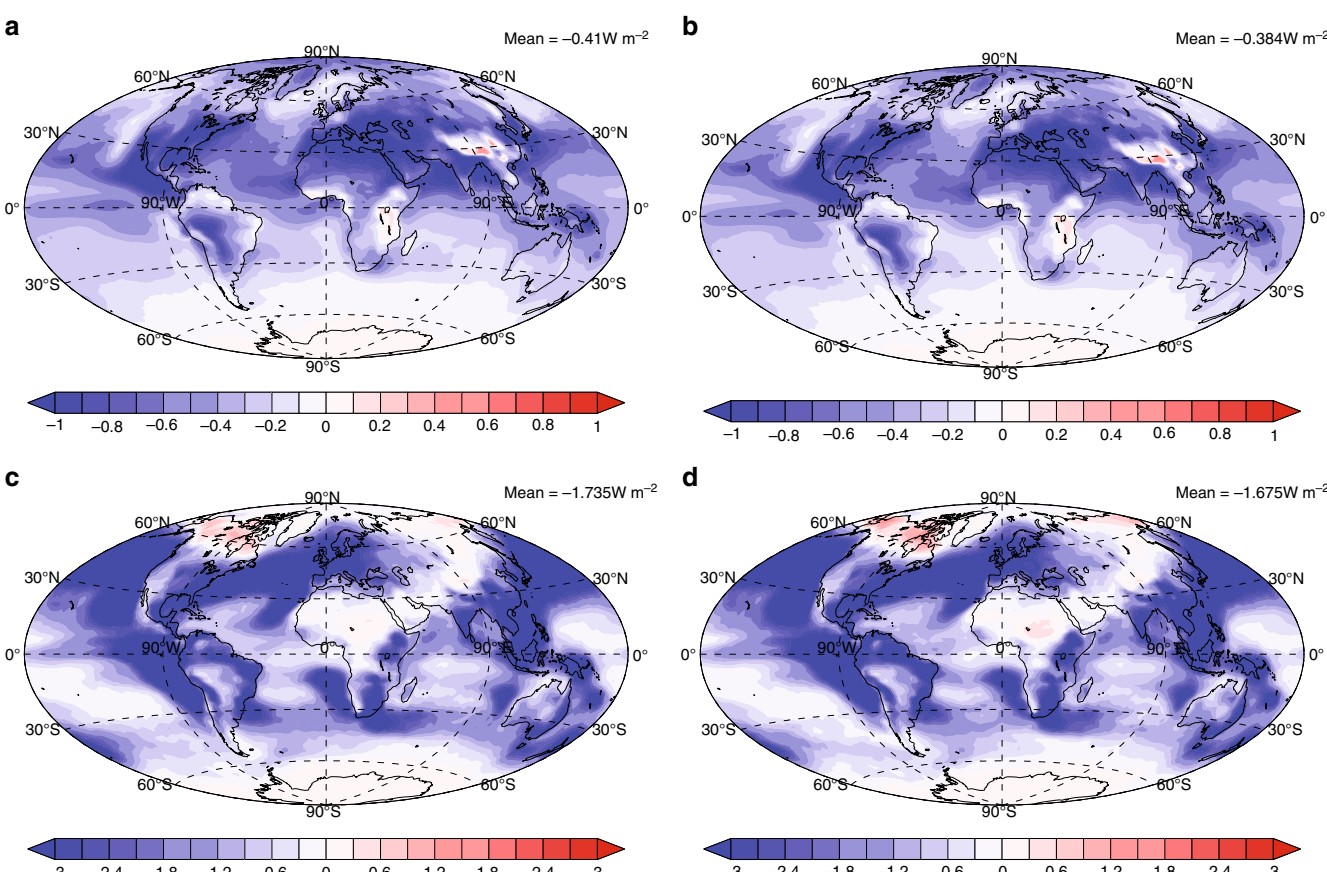

**Fig. 2** The radiative forcing of anthropogenic aerosol with different preindustrial schemes. **a–d** The direct radiative forcing (DRF; **a**, **b**) and indirect radiative forcing (IRF; **c**, **d**) of anthropogenic aerosol with the preindustrial values from PIemi (**a**, **c**) and PIall (**b**, **d**). The global average radiative forcing is shown on the top right of each panel

addition, the large number concentration of newSOA decreases water supersaturations drastically due to enhanced condensation. SOA enhances the global average accumulation mode particle number concentration in the PBL by 22 cm$^{-3}$ (Supplementary Table 1) and the CDNC at the top of water liquid clouds by $1.32 \times 10^6$ m$^{-3}$ (Supplementary Fig. 12a), resulting in an AIE of SOA of $-0.148$ W m$^{-2}$ in the PD (Supplementary Fig. 13a). However, in regions of the NH where the number concentration of sulfate is large, the CDNC at the top of liquid water clouds decreases due to SOA because the maximum supersaturation is significantly reduced by the newSOA and this suppresses many of other particles from being activated to cloud droplets. SOA with organic nucleation results in a positive AIE of SOA in the regions of the NH in the PD (Supplementary Fig. 13a). SOA also lowers the supersaturation in the PI atmosphere, but the CDNC always increases due to SOA because of the low number concentrations of other aerosols. Organic nucleation results in a column number of newSOA in the accumulation mode of $2.8 \times 10^{10}$ and $2.4 \times 10^{10}$ m$^{-2}$ for PIemi and PIall, respectively (Table 1). The SOA in PIemi causes an increase in the CNDC by $9.91 \times 10^6$ m$^{-3}$ at the top of liquid water clouds (Supplementary Fig. 12b), while in PIall the CDNC increases by $9.69 \times 10^6$ m$^{-3}$ due to the smaller number concentration of newSOA (Supplementary Fig. 12c). As a result, the AIE due to SOA in PIall is 8.3% smaller than that in PIemi (Supplementary Fig. 13b, c). However, the column number concentration of new sulfate particles in the Aitken and accumulation modes in PIall increases by 15 and 4.5%, respectively, compared to PIemi due to smaller amounts of sulfuric acid condensed on newSOA in PIall. As a result, climate and land use change enhance the preindustrial value of the AIE of SOA. The indirect radiative forcing (IRF) of anthropogenic aerosol is $-1.675$ W m$^{-2}$ using PIall (Fig. 2c), which is 3.5% smaller than the forcing ($-1.735$ W m$^{-2}$) using PIemi (Fig. 2d). Without organic nucleation, the IRF of anthropogenic aerosol is $-1.878$ W m$^{-2}$ using PIall (Supplementary Fig. 14). Thus, the inclusion of organic nucleation leads to a 11% smaller IRF of anthropogenic aerosol, which is much smaller than the 27% reduction indicated in ref. [12]. Some of this difference is due to our use of a temperature dependence for the nucleation rate. For example, Gordon et al.[12] found a 17% reduction of IRF when they used the same temperature dependence as we use (see also Supplementary S3.1). However, some is due to the fact that the organic nucleation in our model primarily occurs in the upper troposphere so that only 4.3 and 6.0% of the total accumulation mode newSOA occurs in the PBL for the PD and PIall, respectively. In contrast, organic particles mostly form near the surface using the methods in ref. [12], which may overestimate the influence of organic nucleation in the PI atmosphere and thus its influence on the IRF.

**Implications and discussion**. This study demonstrates the importance of climate, land use change, and organic nucleation for the evaluation of anthropogenic aerosol RF. Organic nucleation has a large and complex influence on the aerosol size distribution as well as on the activation of particles to form cloud droplets. Here we show that organic nucleation leads to a 12.5% reduction in the direct plus indirect RF between the PD and PI atmosphere. Moreover, the total RF is further reduced by 4.0% when we include the effects of climate and land use change. Thus, we find a 16% smaller forcing due to anthropogenic aerosols as a result of including organic nucleation, land use change and climate change.

The calculation of SOA and the RF is uncertain in part because of poorly defined chemical formation mechanisms. We currently have limited knowledge of the explicit chemical reactions that lead to the production of extremely low volatility HOMs,

although there should be many ways that these form[19]. Here, we included those HOMs that have been shown to contribute to nucleation in quantum chemical calculations, but it is possible that there are other HOMs that contribute to nucleation and will be identified in the future. The use of different HOMs that lead to nucleation is expected to contribute to a large uncertainty in the RF estimation associated organic nucleation. Most experiments indicated that HOMs are produced through H-shift and peroxy radical autoxidation of monoterpenes[32], although some experiments regarded HOMs as a first-generation product from precursor gases, while others include them after a series of oxidation steps[19,33,34]. Here, we have primarily focused on α-pinene as a precursor to HOM. Exocyclic monoterpenes like β-pinene mostly react with OH and do not form HOMs, so while excluding the production of HOMs from exocyclic monoterpenes adds some uncertainty, it is not likely to be large compared to other uncertainties. The suppression of new particle formation by isoprene has been detected in some field and chamber experiments[35–37]. However, it is not clear how isoprene and its oxidation products may change the oxidation chemistry of terpenes[37]. Including isoprene suppression could reduce the number of new organic particles in the PBL further. The temperature dependence of the nucleation rate used in our model is based on a quantum chemical calculation and could cause some uncertainty in the newSOA number concentration. As shown in Supplementary S3.1, the total RF of anthropogenic aerosol is increased by 5.3% without the temperature dependence. The growth rate of newSOA is another important uncertainty that affects the RF estimate. When the low volatility products formed from isoprene epoxydiol (IEPOX), glyoxal and methyl-glyoxal contribute to the growth of newSOA, the total RF is increased by 12% (see Supplementary S3.2). The emissions of aerosols in the PI atmosphere have large differences depending on different models. For example, Hamilton et al.[38] used several fire emissions data sets to suggest that the cloud albedo forcing might be reduced by 35% when larger fire emissions are used in the PI atmosphere. Larger emissions would also alter our results by adding additional condensation sinks. In addition, uncertainties in the change in land use and vegetation type will also alter the BVOC emissions, leading to an uncertainty in the baseline of SOA in the PI atmosphere[39,40].

## Methods

**CESM/IMPACT model**. We used the Community Earth System Model (CESM) version 1.2.2 coupled with the University of Michigan IMPACT aerosol model with resolution of 1.9° × 2.5° to simulate SOA and five other externally mixed aerosol types. Fifteen aerosol species/types were simulated in the model, including sulfate in three modes (i.e., nucleation (<5 nm), Aitken (5–50 nm), and accumulation (>50 nm)), soot from biomass burning (bSoot, i.e., primary organic aerosol and black carbon from biomass burning), soot from fossil fuel and biofuel burning (fSoot, i.e., primary organic carbon and black carbon), and dust and sea salt, the latter two of which were each carried in four separate bins with varying radii. The basic model description and setup can be found in ref. [41]. An explicit gas phase chemical mechanism for the formation of semi-volatile organic products, including the oxidation of isoprene, α-pinene, and aromatics, is included[42,43]. Lower volatility compounds like oligomers form when semi-volatile organics are incorporated into the aerosol phase. SOA formation from aqueous phase reactions of glyoxal and methylglyoxal and the kinetic uptake of IEPOX, glyoxal, and methylglyoxal were included[44]. The mechanism of organic nitrate formation was updated based on ref. [45]. SOA formed by these gaseous and aqueous phase reactions are distributed to preexisting aerosol[14], resulting in SOA being internally mixed with sulfate, bSoot, fSoot, dust, and sea salt.

The above model was extended by adding the formation of extremely low volatility HOMs from α-pinene oxidation and their nucleation. Diacyl peroxide is suspected to be the one kind of HOM that is able to nucleate based on the experimental evidence[19,46]. The oxidation product of pinanediol was observed in new particle formation experiments[47] and in field measurements of nucleation events[48]. Pinic acid and pinanediol were showed to form new particles through quantum chemistry calculations[49]. Therefore, we include diacyl peroxide, pinic acid, pinanediol, and some oxidation products of pinanediol as HOMs that are able to nucleate in the model. Diacyl peroxide and pinic acid were formed by the

ozonization of α-pinene[19], while pinanediol was formed by the oxidation of the products of the reaction of α-pinene and hydroxyl radical[50]. All HOMs were simulated through explicit gaseous and particle phase reactions according to experimental results[19] and the Master Chemical Mechanism (MCM)[18,20,21]. The concentration of HOMs depends on the emissions of α-pinene, its atmospheric oxidation rate, and temperature. The detailed chemical reactions for the formation of HOMs that were added to the model are listed in the Supplementary Table 3. New particle formation by binary sulfuric acid–water nucleation[51] was extended to include the HET, NON, and ION mechanisms. The rate of HET was parameterized following ref. [52], while the rate of NON and ION were parameterized following ref. [10]. The ionization rate in the model was estimated using the method outlined in ref. [53], which was used to calculate the rate of ION. The organic nucleation rates were adjusted from the temperature of the CLOUD experiments (278 K) to other temperatures by multiplying by $\exp(-(T-278)/10)$[54]. The new particles formed from organic nucleation (newSOA) grew to the Aitken and accumulation mode by condensation of SVOC produced as described in the above references and by condensation of sulfuric acid. NewSOA also coagulated with each other as well as with other preexisting aerosols, at which point they were no longer counted as newSOA. Gas phase HOMs that did not nucleate new particles could condense on newSOA and other aerosols. The newSOA were also removed by dry and wet deposition based on their particle size and hygroscopicity.

**Radiative transfer model**. An off-line radiative transfer model was used to calculate aerosol optical properties and activation to form cloud droplets, and thus to estimate the aerosol direct and indirect radiative effects. The concentration, number, and mixing state of aerosols were taken from that calculated in the IMPACT model. NewSOA particles were regarded as internally mixed particles of organics and sulfate. The refractive indices and hygroscopicity of the internal mixtures within each aerosol type were calculated by volume-weighting of each of the individual aerosol species. Köhler theory together with the soluble fraction of each aerosol type was used to predict the amount of water associated with the aerosol, which was also considered to affect the particle size and refractive indices. The model was introduced in detail by ref. [55].

**Simulation scheme**. The PD scheme used PD emissions, climate, and land use. The PIemi scheme was run with emissions from 1750 but with the climate and land use from the PD. The PIall scheme was run with the same anthropogenic emissions as in PIemi but with climate and land use in the PI. The PD anthropogenic and biomass burning emissions are adopted from those used in Community Atmosphere Model (CAM) with Chemistry for 2000[56] and those from Community Emission Data System (CEDS) for 1750[57]. The emission of BVOC are predicted based on MEGAN 2.1 (Model of Emissions of Gases and Aerosols from Nature, version 2.1)[58] which is embedded in the CESM. The BVOC emission rates depend on the light, temperature, leaf area index (LAI), and fractional coverage of plant functional types (PFT). The monthly mean time-varying LAI and PFT take into account the anthropogenic land use/land cover change for PD (year 2000) and PI (year 1850)[59]. The sea surface temperature and sea ice distributions are prescribed using the HadOIBl dataset in CAM for PD (year 2000) and PI (year 1850)[60]. The concentration of $CO_2$ is set to 367.0 ppm for the PD and 284.7 ppm for the PI. The simulation of climate is determined by sea surface temperatures, the sea ice distribution, and $CO_2$ concentration. Each model simulation was performed for 5 years with a 1-year spin-up. We used 5 years of monthly average aerosol concentrations together with the first year of 4 hourly meteorological conditions to calculate the radiative effects[61,62]. The DRE and AIE of SOA are defined as the difference in radiation at the top of atmosphere (TOA) with and without SOA. The anthropogenic RF was defined as the difference between the radiation at TOA simulated with all aerosols in PD and PIemi or PIall scheme.

**Code availability**. The updated CESM/IMPACT model used in this paper is archived at https://doi.org/10.7302/Z20Z71JP.

## Data availability

The model results for PD, PIall, and PIemi used in this paper is archived at https://doi.org/10.7302/Z20Z71JP. Plots of data and the map on several of the figures were produced using the National Center for Atmospheric Research (NCAR) Command Language (Version 6.5.0) [Software]. (2018). Boulder, Colorado: UCAR/NCAR/CISL/TDD. https://doi.org/10.5065/D6WD3XH5.

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

## Acknowledgements

We are grateful for funding from the NASA ACMAP program under grant number NNX15AE34G as well as an NSF-GEO grant number 1540954. F.Y. acknowledges support from NSF-GEO under grant 1550816. Computer time was provided by the NCAR CISL. The aircraft data in the Amazon were obtained during the ACRIDICON–CHUVA campaign funded by the Max Planck Society, the German Aerospace Center (DLR), FAPESP (São Paulo Research Foundation), and the German Science Foundation (Deutsche Forschungsgemeinschaft, DFG). We thank Jose Jimenez, Rodney Weber, Lynn Russell, Roya Bahreini, Middle Brook, Colette Heald, and their groups for the OC concentration data from aircraft campaigns. We thank Ann Hjell-brekke and the EMEP network for the surface OC concentration and surface aerosol number concentration in Europe. We thank the IMPROVE network for the surface OC concentration in the United States.

## Author contributions

J.Z. and J.E.P. conceived and designed the research. J.Z., F.Y., and S.S. developed the model. M.O.A. and H.C. provided observation data. J.Z. and J.E.P. wrote the manuscript.

## Additional information

**Competing interests:** The authors declare no competing interests.

