## [peer review file · Nature Communications]

Reviewer #1 (Remarks to the Author):

Review of "Organic aerosol nucleation, climate and land use change: Decrease in radiative forcing".

Hamish Gordon, 6/8/18

This article explores the implications for atmospheric particle number concentrations and radiative balance of the mechanisms for new particle formation (NPF) from highly oxidised organic molecules (HOMs) alone, and HOMs with sulphuric acid, first quantified at the CLOUD experiment. The authors produce HOM-like molecules in a different way to the previous implementation of the mechanisms in atmospheric models (Gordon et al 2016, Riccobono et al 2014) and, after some other tweaking, they produce improved agreement of their model with observations in the Amazon compared to Gordon et al 2016. This difficult region to model is where this mechanism should be visible, and where the previous implementation produced results in conflict with observations. I do think the authors should show how their model compares to measurements elsewhere as well, though.

The key advances made in the manuscript are the demonstration that good agreement with observations can be obtained in a model including pure biogenic NPF, the calculation of the effect of land use change on organic NPF, the calculation of the effect of organic NPF on pre-industrial and present-day supersaturations in water clouds (at low model resolution), and the direct effect of organic NPF. The chemical mechanism is different (better in some ways, worse in others), and it is obviously valuable to explore possible alternatives to previous work. The model used is more sophisticated than the model used by Riccobono et al (2014) and Gordon et al (2016).

The manuscript addresses an important topic, and I believe that, taken as a whole, it will be a sufficiently significant advance on previous work to be appropriate for Nature Communications once my comments below are addressed. It is well written, previous literature is discussed appropriately, and I was pleased to be asked to review it. My 'major' comments should not require a huge amount of extra work to address.

Major comments

1. NPF processes and model uncertainties

As well as implementing pure organic new particle formation in a different (and more sophisticated) host model, the authors make two key changes to the new particle formation process in their model compared to Gordon et al (2016). The first is to use an explicit chemical mechanism for HOM production which does not involve autoxidation, instead of trying to simulate autoxidation in the model. The second is to use a temperature dependence for the NPF rates as their baseline, which was introduced as a sensitivity study by Gordon et al (and was only ever intended as a sensitivity study, as it has no physical basis).

1.1 Chemical mechanism

Line 293: "Most experiments indicated that HOMs are produced through H-shift and peroxy radical autoxidation". The basic postulate presented by the authors at the process level seems to be that this doesn't happen in the atmosphere.

This hypothesis seems quite unlikely to be strictly true, given that the chamber and field-measured mass spectra for HOMs are quite similar (Schobesberger et al 2013). In terms of individual molecules, the chamber and field spectra are certainly more similar to each other than they are similar to the mass spectrum of HOMs the authors produce using the MCM model.

That said, the yield of HOMs via the autoxidation of α -pinene could well be quite different to the yield we measured in the CLOUD chamber and used in Gordon et al, 2016. NO_x, RO₂ (e.g. from isoprene), and HO₂ are all possible candidates to suppress this yield, as the authors also point out. Furthermore, there's nothing to guarantee the molecules that form particles at 5°C in the CLOUD

chamber are the same as those which form particles in the upper troposphere either, which is where, in this paper, most pure biogenic NPF is happening. Indeed, one would expect higher volatility HOMs such as those the authors produce to be able to stick together at low upper tropospheric temperatures. Consequently, not simulating autoxidation might be a reasonable approximation, in the end, when one considers that the alternative, of trying to simulate it using yields from pure alpha-pinene ozonolysis experiments, is also certainly not ideal.

Ultimately there are two approaches here: here, the authors use chemical reactions that are explicitly modelled in MCM and therefore should happen in the atmosphere, to produce molecules which may or may not form new particles. Before, we used a representation of chemical reactions that happen in chamber experiments but are not guaranteed to happen at the same rate in the atmosphere, to make molecules that we could be more confident can form particles. The approaches are different –it is too early to say which is better - and it is valuable that the authors have explored the one that we didn't explore already.

To make things clearer, I think the authors should emphasise explicitly somewhere that they do not simulate autoxidation. Also, I think it would be useful to show the effect of the new chemical mechanism separately from the effect of the temperature dependence the authors introduce.

1.2. Temperature dependence

In the methods, the authors say "The organic nucleation rates were multiplied by $\exp(-(T-278)/10)$. (Dunne et al, 2016)" This temperature dependence is important, and will substantially affect the results of this study. When we used this same T dependence for a sensitivity study in Gordon et al (2016), we found that the cloud albedo forcing was reduced to +0.14Wm⁻² from 0.22Wm⁻². The 0.14Wm⁻² (a 17% reduction in forcing, rather than a 27%) is stated in the main text of Gordon et al (2016) and is much more similar to the results of the present manuscript, although in Gordon et al it is only the forcing from pure organic NPF; in this manuscript the forcing effect of H₂SO₄+HOM NPF is also included.

Unfortunately I have no evidence for this temperature dependence. It is only an educated guess. On page 31 of the Dunne et al SI I wrote that it has "no physical basis", maybe looking back, I should have restated this in the Table S7 caption (sorry!). We didn't have any experimental data with which to get this T dependence at the time.

The authors should state explicitly that the T dependence that they used is not supported by published experimental evidence and they should explore the sensitivity of their results to the temperature dependence with new simulations. Maybe also worth referring to Yu et al (ACP 2017). This useful study considers the possible T dependence of the NPF rate, though not of the HOM production mechanism (if one assumes autoxidation is the main mechanism for HOM formation, it has a strong T dependence which would counteract to some unknown extent the T dependence of the NPF rate).

2. Particle growth

L127: "The global average mass of sulfuric acid in the nucleation and Aitken mode particles in newSOA is larger than that of organics. Thus, condensed sulfuric acid and water are responsible for most of the growth of small newSOA particles after organics drive the formation of new particles, consistent with some laboratory and field measurements (27,28)."

Put crudely, if a molecule has low enough volatility to form new particles, it should also be able to grow them. Of course the dependences on vapour concentration can be different, but ultimately I think that if NPF is dominated by organics and not H₂SO₄, growth should also be dominated by organics and not H₂SO₄, with H₂SO₄ making only a minor contribution. The only situation (apart from strange chemical reactions) in which this would not be true is if the nucleating organics had lower average volatility than H₂SO₄ and H₂SO₄ was more abundant in the gas phase. This seems unlikely, though not impossible, and the possibility is not supported by the references cited by the authors. The reference 27 documents chamber measurements of growth rates due to H₂SO₄, and reference 28 is atmospheric observations of organic NPF. Neither of these papers say that newly formed organic clusters grow mostly by condensation of H₂SO₄. On the other hand, Lawler et al (GRL 2018) show that 90% of the mass of newly formed particles in Hyytiälä of around 40nm in

diameter is organic. I think the statement of the authors points to an inconsistency between the formation of nucleating organics in their model and the formation of the organics that grow the particles (or perhaps the NPF rate at low temperatures is too high compared to the binary H₂SO₄ formation rate). This possibility should at least be acknowledged in the text. Further, at line 132, the concentration of SVOC over the Amazon is surely much higher than the concentration of H₂SO₄, not really comparable.

The authors acknowledge implicitly that this has large consequences for their results, at line 166 and 174. It seems that about half the increase in newSOA number concentrations in PD compared to PI is due to increased NPF rates and half is due to condensation of H₂SO₄. The increased newSOA in the PD compared to the PI may be exaggerated if organics are insufficiently able to grow nucleation-mode particles (especially in the PI). This would suppress the PD-PI indirect forcing. Conversely, too high organic formation rates might exaggerate the PD-PI indirect forcing. Ideally, the authors should do a sensitivity simulation, along the lines of allowing more SVOC to condense onto smaller particles, and some discussion of this possible shortcoming would also be helpful.

3. Model evaluation

While it is clear from comparing Fig.1 and SI Fig 2 that the model does a reasonable job of reproducing observations in the very challenging Amazon environment, the comparison should be made more precise by using the radii of the Aitken mode to cut off the simulated size distribution at 10nm and 20nm as appropriate for an explicit comparison. At the moment, the simulated number concentration is clearly higher than the observed number concentration, but this may not be true if the particles below the CPC cut off size are excluded. Then I think the authors should put the model and measurement data on the same plot in the main text.

The authors show that their model reproduces the Amazon, where organic NPF is not seen. However, what about places where NPF is observed to be much higher in the boundary layer than binary H₂SO₄-H₂O-ion NPF would predict, like Hyytiälä in Finland, or various places in the USA (e.g. Yu et al, ACP 2015)? Does the model reproduce NPF rates observed at Hyytiälä and/or other polluted boundary layer sites strongly influenced by BVOCs? And what about (e.g.) northern Finland, Canada or Siberia, where one might also expect pure organic NPF?

Minor comments

Abstract:

"Organic nucleation" (also line 53) is a bit vague; please specify "nucleation involving either organics alone or organics with sulphuric acid" once in the abstract.

Towards the end it is not clear whether climate and land use change cause an enhancement of 3% on the DRF independently of organic NPF or only because of organic NPF.

Line 56 "BVOC nucleation" is also unclear – this was NPF of BVOCs and H₂SO₄.

Line 62: See major comment: please could the authors mention here that the assumption that HOM yields are instantaneous is, to a very good approximation, what is observed in chamber studies, and in particular in the chamber studies from which their NPF parameterisation is derived (e.g. Ehn et al, 2014, Kirkby et al, 2016).

Exocyclic monoterpenes react differently with ozone and OH than endocyclic monoterpenes, so about 40% of monoterpenes (see SI of Jokinen PNAS 2015) mostly react with OH and mostly don't form HOMs. We tried to replicate this in Gordon et al (2016) and it did make quite a sizeable difference compared to just using alpha-pinene, though it is by no means the most important uncertainty. This might be worth discussing in your "implications".

Line 117: it would be worth comparing to Fig 5 in Gordon et al JGR 2017 here. The relative fractions of NPF from ION and HET are quite different. This may be related to the altitude range used. In SI Fig 4 it would also be instructive to add a panel with the nucleation rate from sulphuric acid binary NPF?

Line 248: "Organic nucleation results in accumulation mode number concentrations of newSOA of 1.9 and 1.4cm⁻³." These very small numbers are a bit surprising, given the 10x larger increase in CDNC and also the substantial change in supersaturation. Presumably this means some of the Aitken mode newSOA is activating, or the activation doesn't happen in the PBL. It would be helpful to know the mean diameters of the modes involved and the indirect effect of all aerosol to understand this better. The CDNC changes also seem small given the large AIE. Scott et al (ACP 2014, Table 2) found that a PI change of 35cm⁻³ in CDNC or 38% led to an AIE of -0.95Wm⁻². This is a 30% lower AIE than your result, for a 3x higher change in CDNC. It would be good to try to understand what is driving this, though I understand it may be difficult and beyond the scope of the paper. Please could you state your relative increase in global mean CDNC? This may be enough to give us some idea.

Methods:

Please state the horizontal and vertical resolution of the model.

Please cite the source for the binary H₂SO₄ NPF mechanism.

Do you track the sink of newSOA due to coagulation? I would expect this is important compared to dry/wet deposition in the Amazon biomass burning (dry) season.

Please specify the size ranges (particle diameters) that the nucleation, Aitken and accumulation modes are designed to cover.

Please state the hygroscopicity (Kappa or van t'Hoff factor) assumed for SOA.

Please state the origin of the updraft speeds used in the offline activation calculation; I couldn't immediately see this in ref. 43.

The overall formation rates seem very high in the Amazon. Are the formation rates at very low temperatures allowed to exceed the kinetic limit for HOM-HOM intermolecular collisions?

Are the radiative effects of newSOA calculated by taking the difference between a simulation with the NPF mechanism switched on and a simulation with the NPF mechanism switched off. Or are these effects all calculated from the same simulation, by ignoring and then not ignoring the newSOA modes in the activation/radiative transfer scheme?

Fig. S2 caption: "cited from"->"reproduced from". The figure is currently a screenshot from Andreae et al; incorporating this data into Fig 1 may make the permissions requirements easier?

Fig S4, S8: Please use a logarithmic colour scale.

Fig S5 Please state the units more clearly, and explain them in the caption.

Reviewer #2 (Remarks to the Author):

Zhu et al., Organic aerosol nucleation, climate and land use change: Decrease in radiative forcing

Main claims/findings of the paper are that organic nucleation is an important source of aerosols in the pristine continental areas and during the preindustrial period, and that the increased emissions of α -pinene and its oxidation products (nucleating and forming SOA) are enhancing due to the climate and land use changes. Together these events: organic nucleation, land use and climate, lead to an overall 15 % reduction in radiative forcing when applied together with present day anthropogenic aerosol in the tropics. This paper also demonstrates reproductions of vertical profile of condensation nuclei over the Amazon forest.

I only have more general comments to this work rather than going into details:

- The subject of the study is very complex and the study seems novel at least considering most parts of the work. I'm pleased to see that the effect of climate and land use is included in this work with particle formation happening above the planetary boundary layer. As organic (pure biogenic) nucleation has only been observed and published a few times (Kirkby et al., Nature = lab study and Bianchi et al., Science = field study), models are not accounting it in yet. Even if I consider the methods and results to be novel, I feel that there are too many misinterpretations and misunderstanding with the recent experimental work done in the field of nucleation studies and HOMs. I also find referencing to previous work inadequate and somewhat misleading, especially considering the referencing of experimental publications. Thus, I cannot advise publication in the current form.

- I found that it was difficult to understand if all the work is concerning the vertically integrated values or upper troposphere/PBL conditions only. This should be clarified throughout the manuscript.

- Applying experimental information: The paper claims that the Amazon is the most important region producing new organic particles and has the highest organic nucleation rate primarily because of the high emissions of α -pinene and the high production of HOMs from α -pinene oxidation. My knowledge is that the Amazon is known for its extremely high concentrations and emissions of isoprene, not α -pinene. Even though both α -pinene and isoprene emissions are estimated to increase due to land use and temperature, isoprene is still clearly dominating the BVOCs in the Amazonas. It should be noted in the manuscript that the large fraction of SOA forming HOMs have very different formation pathways, isoprene is mostly oxidized by OH-radicals to form HOM and monoterpenes produce HOM from ozonolysis (e.g. Ehn et al., 2014), OH-radicals (E.g. Berndt et al., 2016) and NO₃-radicals (E.g. Ayres et al., 2015, Yan et al., 2016). Major changes in the oxidation pathways could cause very different results in nucleation, SOA formation and load. Lots of HOM related work, production mechanisms, yields of different HOMs from various monoterpenes, sesquiterpenes and isoprene (SOAS campaign, Alabama) has been published since the initial Ehn et al, 2014 paper.

Response to the reviewers' comments on “Organic aerosol nucleation, climate, and land use change: Decrease in radiative forcing”

Jialei Zhu¹, Joyce E. Penner^{1*}, Fangqun Yu², Sanford Sillman¹, Meinrat O. Andreae^{3,4}, Hugh Coe⁵

1 Department of Climate and Space Sciences and Engineering, University of Michigan, Ann Arbor, Michigan 48109, USA

2 Atmospheric Sciences Research Center, State University of New York at Albany, Albany, New York 12203, USA

3 Biogeochemistry Department, Max Planck Institute for Chemistry, Mainz, Germany

4 Scripps Institution of Oceanography, University of California San Diego, La Jolla, CA 92093, USA

5 School of Earth and Environmental Sciences, University of Manchester, Manchester M13 9PL, UK

Corresponding Author:

Joyce E. Penner

Address: Space Research Building, University of Michigan, 2455 Hayward Street, Ann Arbor, Michigan 48109, USA

Telephone Number: (734) 936-0519

Email: penner@umich.edu

We thank editor and two reviewers very much for dedicating their time to read our manuscript and present important comments. We carefully studied these comments and revised the manuscript widely. In summary, we updated the chemical mechanism in our model to reduce uncertainty based on reviews' comments. We also added model validations and sensitivity experiments in the supplementary. Replies to these comments are listed point by point as below.

Reviewers' comments:

Reviewer #1:

Review of “Organic aerosol nucleation, climate and land use change: Decrease in radiative forcing”.

Hamish Gordon, 6/8/18

This article explores the implications for atmospheric particle number concentrations and radiative balance of the mechanisms for new particle formation (NPF) from highly oxidised organic molecules (HOMs) alone, and HOMs with sulphuric acid, first quantified at the CLOUD experiment. The authors produce HOM-like molecules in a different way to the previous implementation of the mechanisms in atmospheric models (Gordon et al 2016, Riccobono et al 2014) and, after some other tweaking, they produce improved agreement of their model with observations in the Amazon compared to Gordon et al 2016. This difficult region to model is where this mechanism should be visible, and where the previous implementation produced results in conflict with observations. I do think the authors should show how their model compares to measurements elsewhere as well, though.

Re: We have added the comparison of surface OC concentration, vertical profile of OC concentration and surface aerosol number concentration between simulation and observation at a number of sites in the supplementary information Section S1.

The key advances made in the manuscript are the demonstration that good agreement with observations can be obtained in a model including pure biogenic NPF, the calculation of the effect of land use change on organic NPF, the calculation of the effect of organic NPF on pre-industrial and present-day supersaturations in water clouds (at low model resolution), and the direct effect of organic NPF. The chemical mechanism is different (better in some ways, worse in others), and it is obviously valuable to explore possible alternatives to previous work. The model used is more sophisticated than the model used by Riccobono et al (2014) and Gordon et al (2016).

The manuscript addresses an important topic, and I believe that, taken as a whole, it will be a sufficiently significant advance on previous work to be appropriate for Nature Communications once my comments below are addressed. It is well written, previous literature is discussed appropriately, and I was pleased to be asked to review it. My ‘major’ comments should not require a huge amount of extra work to address.

Major comments

1. NPF processes and model uncertainties

As well as implementing pure organic new particle formation in a different (and more sophisticated) host model, the authors make two key changes to the new

particle formation process in their model compared to Gordon et al (2016). The first is to use an explicit chemical mechanism for HOM production which does not involve autoxidation, instead of trying to simulate autoxidation in the model. The second is to use a temperature dependence for the NPF rates as their baseline, which was introduced as a sensitivity study by Gordon et al (and was only ever intended as a sensitivity study, as it has no physical basis).

1.1 Chemical mechanism

Line 293: “Most experiments indicated that HOMs are produced through H-shift and peroxy radical autoxidation”. The basic postulate presented by the authors at the process level seems to be that this doesn’t happen in the atmosphere.

This hypothesis seems quite unlikely to be strictly true, given that the chamber and field-measured mass spectra for HOMs are quite similar (Schobesberger et al 2013). In terms of individual molecules, the chamber and field spectra are certainly more similar to each other than they are similar to the mass spectrum of HOMs the authors produce using the MCM model.

That said, the yield of HOMs via the autoxidation of α -pinene could well be quite different to the yield we measured in the CLOUD chamber and used in Gordon et al, 2016. NO_x , RO_2 (e.g. from isoprene), and HO_2 are all possible candidates to suppress this yield, as the authors also point out. Furthermore, there’s nothing to guarantee the molecules that form particles at 5°C in the CLOUD chamber are the same as those which form particles in the upper troposphere either, which is where, in this paper, most pure biogenic NPF is happening. Indeed, one would expect higher volatility HOMs such as those the authors produce to be able to stick

together at low upper tropospheric temperatures. Consequently, not simulating autoxidation might be a reasonable approximation, in the end, when one considers that the alternative, of trying to simulate it using yields from pure alpha-pinene ozonolysis experiments, is also certainly not ideal.

Ultimately there are two approaches here: here, the authors use chemical reactions that are explicitly modelled in MCM and therefore should happen in the atmosphere, to produce molecules which may or may not form new particles. Before, we used a representation of chemical reactions that happen in chamber experiments but are not guaranteed to happen at the same rate in the atmosphere, to make molecules that we could be more confident can form particles. The approaches are different –it is too early to say which is better - and it is valuable that the authors have explored the one that we didn't explore already.

To make things clearer, I think the authors should emphasize explicitly somewhere that they do not simulate autoxidation. Also, I think it would be useful to show the effect of the new chemical mechanism separately from the effect of the temperature dependence the authors introduce.

Re: We emphasized our explicit chemical mechanism is quite different with the assumption of fast autoxidation used in your studies (around Line 65). We also noted that your model applied HOMs yield and nucleation parameter purely based on CLOUD experiment, which may not reflect what happens in the atmosphere exactly (around Line 69). We added a discussion of the effects of using the temperature dependence of new particle formation rates in supplementary information Section S3.1 for a result from a sensitivity test without temperature dependence. We found the aerosol number concentration will peak around 1.5 km

in the Amazon when the model without temperature dependence is used. At the same time, there is still a small peak of aerosol number concentration in the upper troposphere around 12 km in the Amazon. As a result, the sensitivity experiment results in a 5.3% increase of total radiative forcing of anthropogenic aerosol compared to the result calculated in the base model in the main text.

1.2. Temperature dependence

In the methods, the authors say “The organic nucleation rates were multiplied by $\exp(-(T-278)/10)$. (Dunne et al, 2016)” This temperature dependence is important, and will substantially affect the results of this study. When we used this same T dependence for a sensitivity study in Gordon et al (2016), we found that the cloud albedo forcing was reduced to $+0.14\text{Wm}^{-2}$ from 0.22Wm^{-2} . The 0.14Wm^{-2} (a 17% reduction in forcing, rather than a 27%) is stated in the main text of Gordon et al (2016) and is much more similar to the results of the present manuscript, although in Gordon et al it is only the forcing from pure organic NPF; in this manuscript the forcing effect of $\text{H}_2\text{SO}_4+\text{HOM}$ NPF is also included.

Unfortunately I have no evidence for this temperature dependence. It is only an educated guess. On page 31 of the Dunne et al SI I wrote that it has “no physical basis”, maybe looking back, I should have restated this in the Table S7 caption (sorry!). We didn’t have any experimental data with which to get this T dependence at the time.

The authors should state explicitly that the T dependence that they used is not supported by published experimental evidence and they should explore the sensitivity of their results to the temperature dependence with new simulations.

Maybe also worth referring to Yu et al (ACP 2017). This useful study considers the possible T dependence of the NPF rate, though not of the HOM production mechanism (if one assumes autoxidation is the main mechanism for HOM formation, it has a strong T dependence which would counteract to some unknown extent the T dependence of the NPF rate).

Re: We added a discussion of uncertainty of temperature dependence in supplementary S3.1 for the result from the sensitivity test without temperature dependence for NPF. In that section, we also noted the temperature dependence applied in base model in the main text was based on the quantum chemical calculations of cluster binding energies and lack of published experimental evidence till now, so the temperature dependence may have uncertainty.

2. Particle growth

L127: “The global average mass of sulfuric acid in the nucleation and Aitken mode particles in newSOA is larger than that of organics. Thus, condensed sulfuric acid and water are responsible for most of the growth of small newSOA particles after organics drive the formation of new particles, consistent with some laboratory and field measurements (27,28).”

Put crudely, if a molecule has low enough volatility to form new particles, it should also be able to grow them. Of course the dependences on vapour concentration can be different, but ultimately I think that if NPF is dominated by organics and not H₂SO₄, growth should also be dominated by organics and not H₂SO₄, with H₂SO₄ making only a minor contribution. The only situation (apart from strange chemical reactions) in which this would not be true is if the nucleating organics had lower average volatility than H₂SO₄ and H₂SO₄ was

more abundant in the gas phase. This seems unlikely, though not impossible, and the possibility is not supported by the references cited by the authors. The reference 27 documents chamber measurements of growth rates due to H₂SO₄, and reference 28 is atmospheric observations of organic NPF. Neither of these papers say that newly formed organic clusters grow mostly by condensation of H₂SO₄. On the other hand, Lawler et al (GRL 2018) show that 90% of the mass of newly formed particles in Hyytiälä of around 40nm in diameter is organic. I think the statement of the authors points to an inconsistency between the formation of nucleating organics in their model and the formation of the organics that grow the particles (or perhaps the NPF rate at low temperatures is too high compared to the binary H₂SO₄ formation rate). This possibility should at least be acknowledged in the text. Further, at line 132, the concentration of SVOC over the Amazon is surely much higher than the concentration of H₂SO₄, not really comparable.

The authors acknowledge implicitly that this has large consequences for their results, at line 166 and 174. It seems that about half the increase in newSOA number concentrations in PD compared to PI is due to increased NPF rates and half is due to condensation of H₂SO₄. The increased newSOA in the PD compared to the PI may be exaggerated if organics are insufficiently able to grow nucleation-mode particles (especially in the PI). This would suppress the PD-PI indirect forcing. Conversely, too high organic formation rates might exaggerate the PD-PI indirect forcing. Ideally, the authors should do a sensitivity simulation, along the lines of allowing more SVOC to condense onto smaller particles, and some discussion of this possible shortcoming would also be helpful.

Re: The HOMs condensed on the new particles is able to grow them up. However, the concentration of HOMs is much less than the concentration of H₂SO₄ even if in the tropics, so that HOMs is not able to grow them as quickly as H₂SO₄ does. There may be much higher concentration of HOMs in the real atmosphere, but we currently have limited knowledge of the explicit chemical reactions that lead to the production of extremely low volatility HOMs. In our model, the growth of newSOA by organics is dominated by the partitioning of SVOC. Notably most of the partitioning of SVOC takes place on particles with a large mass of pre-existing organics rather than on the smaller particles. As the result, organics don't have a large contribution to the growth of very small newSOA particles, but contribute a lot to the growth of larger newSOA particles. Our model results indicate that the mass of organics in tropics always accounts for >80% of the total mass of newSOA particle with diameter >50nm, which agrees with the measurements in Hyytiala. We added a sensitivity test to examine the effect of different mechanisms for uptake of organics on newSOA in supplementary information section S3.2. We assumed IEPOX, glyoxal and methylglyoxal are also able to condense on newSOA, whereas they are only taken up by new sulfate particles in the base model in the main text. In this sensitivity test, the low-volatility products formed from IEPOX, glyoxal and methylglyoxal take part in the growth of newSOA. As a result, organics dominate the growth of newSOA in the tropics. The mass concentration of organics on newSOA in each mode is larger by a factor of 4-7 than that of sulfuric acid in the tropics. For Line 132 (now around Line 140), we meant the concentration of organics condensed on newSOA is comparable to that of sulfuric acid on newSOA, but not the concentration of SVOC and sulfuric acid in the atmosphere of Amazon. We have changed that sentence to “the concentration of organics on the newSOA is comparable to that of sulfuric acid”.

3. Model evaluation

While it is clear from comparing Fig.1 and SI Fig 2 that the model does a reasonable job of reproducing observations in the very challenging Amazon environment, the comparison should be made more precise by using the radii of the Aitken mode to cut off the simulated size distribution at 10nm and 20nm as appropriate for an explicit comparison. At the moment, the simulated number concentration is clearly higher than the observed number concentration, but this may not be true if the particles below the CPC cut off size are excluded. Then I think the authors should put the model and measurement data on the same plot in the main text.

The authors show that their model reproduces the Amazon, where organic NPF is not seen. However, what about places where NPF is observed to be much higher in the boundary layer than binary H₂SO₄-H₂O-ion NPF would predict, like Hyytiälä in Finland, or various places in the USA (e.g. Yu et al, ACP 2015)? Does the model reproduce NPF rates observed at Hyytiälä and/or other polluted boundary layer sites strongly influenced by BVOCs? And what about (e.g.) northern Finland, Canada or Siberia, where one might also expect pure organic NPF?

Re: We have merged the Figure 1a and Figure S2 into the same figure (Figure 1a) using the same size cutoff. Moreover, we added an evaluation of the model by comparing to observations of surface OC concentration, the vertical profile of OC concentration and surface aerosol number concentration in the supplementary information section S1. Our model does not show as large a contribution of organic nucleation in the Finland, Canada or Siberia as in the Amazon. The concentration

of HOMs in Finland, Canada is also much less than that in the Amazon, even in the summer. The correlation coefficient between monthly simulated number concentration of newSOA and observed aerosol number concentration at the three sites in Finland and one site in Canada are all higher than 0.6, which indicates the model is able to reproduce the pattern associated with the seasonal variation of aerosol number concentration in regions strongly influenced by BVOCs.

Minor comments

Abstract:

“Organic nucleation” (also line 53) is a bit vague; please specify “nucleation involving either organics alone or organics with sulphuric acid” once in the abstract.

Re: We have specified with “pure organic nucleation and heteromolecular nucleation of sulfuric acid and organics” in the beginning of abstract. Also, added “involving either pure organics or organics with sulfuric acid” around L58

Towards the end it is not clear whether climate and land use change cause an enhancement of 3% on the DRF independently of organic NPF or only because of organic NPF.

Re: In the latest version, the DRF is increased by 18% as a result of only organic nucleation. Climate and land use change alone cause an enhancement of 6.3% on the DRF. The schemes for the difference between with and without climate and land use change in the PI both include organic nucleation. We have changed the

discussion of these two differences in that section (around L222~L231). The description is hopefully much clearer now.

Line 56 “BVOC nucleation” is also unclear – this was NPF of BVOCs and H₂SO₄.

Re: “new particle formation from BVOC and sulfuric acid” has been added around L61.

Line 62: See major comment: please could the authors mention here that the assumption that HOM yields are instantaneous is, to a very good approximation, what is observed in chamber studies, and in particular in the chamber studies from which their NPF parameterisation is derived (e.g. Ehn et al, 2014, Kirkby et al, 2016).

Re: We have changed that sentence to “previous model studies applied empirical or semi-empirical fixed and instantaneous HOM yields. These studies assumed that autooxidation occurs rapidly and produces HOMs after the initial oxidation step based on the CLOUD project experiments.” around L66-68.

Exocyclic monoterpenes react differently with ozone and OH than endocyclic monoterpenes, so about 40% of monoterpenes (see SI of Jokinen PNAS 2015) mostly react with OH and mostly don’t form HOMs. We tried to replicate this in Gordon et al (2016) and it did make quite a sizeable difference compared to just using alpha-pinene, though it is by no means the most important uncertainty. This might be worth discussing in your “implications”.

Re: Thank you for suggestion. We have added “Exocyclic monoterpenes like β -pinene mostly react with OH and do not form HOMs, so while excluding the

production of HOMs from exocyclic monoterpenes adds some uncertainty, it is not likely to be large compared to other uncertainties.” in the implications and discussion around L298.

Line 117: it would be worth comparing to Fig 5 in Gordon et al JGR 2017 here. The relative fractions of NPF from ION and HET are quite different. This may be related to the altitude range used. In SI Fig 4 it would also be instructive to add a panel with the nucleation rate from sulphuric acid binary NPF?

Re: We have added the comparison of nucleation rate between these two studies in the supplementary S2. The fraction of new particle formation from sulfuric acid is 10% higher in our model than that in the Gordon et al. model. That is probably because there is a large fraction of organic nucleation occurring in the middle and upper troposphere in our model while organic nucleation always occurs within the PBL in the Gordon et al. study. The fraction of new particle formation from heteromolecular nucleation of sulfuric acid and organics (HET) is smaller in our model, which may be caused by our neglect of the ion-induced HET pathway. As a result, more HOMs would take part in the ION pathway to form new organic particles in our model than the Gordon et al. study, leading to a higher fraction of new particle formation from ION in our result.

This study		Gordon et al. (2017)	
Pathway	Fraction	Pathway	Fraction
ION	23.2%	org-ion	4.1%
NON	0.6%	Neutral organic	0.4%
HET	17.8%	SA-org	47.0%
		SA-org-ion	
H ₂ SO ₄ +H ₂ O	58.4%	SA-ion	48.5%
		SA-NH ₃	
		SA-NH ₃ -ion	

Line 248: “Organic nucleation results in accumulation mode number concentrations of newSOA of 1.9 and 1.4cm⁻³.” These very small numbers are a bit surprising, given the 10x larger increase in CDNC and also the substantial change in supersaturation. Presumably this means some of the Aitken mode newSOA is activating, or the activation doesn’t happen in the PBL. It would be helpful to know the mean diameters of the modes involved and the indirect effect of all aerosol to understand this better. The CDNC changes also seem small given the large AIE. Scott et al (ACP 2014, Table 2) found that a PI change of 35cm⁻³ in CDNC or 38% led to an AIE of -0.95Wm⁻². This is a 30% lower AIE than your result, for a 3x higher change in CDNC. It would be good to try to understand what is driving this, though I understand it may be difficult and beyond the scope of the paper. Please could you state your relative increase in global mean CDNC? This may be enough to give us some idea.

Re: In the latest version, the CDNC is changed by 9.69 cm⁻³ leading to AIE of -0.308 W m⁻² due to SOA in PIall. This rate of CDNC to AIE is a much better fit to Scott et al. 2014, but the AIE due to SOA is smaller than that shown in Scott et al. 2014. The number concentration of newSOA in the accumulation mode within the PBL is 2.9 cm⁻³ in PIall, so we still believe there are some Aitken mode newSOA particles that are activated and some newSOA that is activated above the PBL.

Methods:

Please state the horizontal and vertical resolution of the model.

Re: we have added “with resolution of 1.9°×2.5°” around L309 in the Methods.

Please cite the source for the binary H₂SO₄ NPF mechanism.

Re: The mechanism for binary sulfuric acid-water nucleation used in our model is based on Vehkamaeki et al. (2002). This has been added to the methods section (around Line 341).

Do you track the sink of newSOA due to coagulation? I would expect this is important compared to dry/wet deposition in the Amazon biomass burning (dry) season.

Re: Yes, we calculated the coagulation of newSOA in the model, but we did not keep track of the totals. Our simulation results are able to reproduce the peak of bOC in the Amazon during the dry season (July-September). The coagulation should be important sink of newSOA when the aerosol number concentration is high.

Please specify the size ranges (particle diameters) that the nucleation, Aitken and accumulation modes are designed to cover.

Re: We added the size ranges around L312, that is “nucleation (<5nm), Aitken (5-50nm) and accumulation (>50nm)”.

Please state the hygroscopicity (Kappa or van t’Hoff factor) assumed for SOA.

Re: We assumed a Kappa factor of 0.14 to describe the hygroscopicity of SOA. When sulfuric acid is coated on newSOA or SOA is internally mixed with other

aerosols, the Kappa factor of the mixed particle is calculated according to the volume fraction of composition and the individual constituent Kappa factors.

Please state the origin of the updraft speeds used in the offline activation calculation; I couldn't immediately see this in ref. 43.

Re: The updraft speeds are calculated from the eddy diffusivity of the CAM model refer to Wang et al. 2009.

The overall formation rates seem very high in the Amazon. Are the formation rates at very low temperatures allowed to exceed the kinetic limit for HOM-HOM intermolecular collisions?

Re: The kinetic limit collision coefficient is calculated as $\sim 3 \times 10^{-10} \text{ cm}^3 \text{ s}^{-1}$ which, for the maximum nucleation rate, results in a kinetically limited nucleation rate of $10^6 \text{ cm}^{-3} \text{ s}^{-1}$ when the temperature is 200K and the concentration of HOMs is highest. Thus, the kinetic limit nucleation rate is much higher than the maximum nucleation rate ($\sim 2 \text{ cm}^{-3} \text{ s}^{-1}$) in our simulation.

Are the radiative effects of newSOA calculated by taking the difference between a simulation with the NPF mechanism switched on and a simulation with the NPF mechanism switched off. Or are these effects all calculated from the same simulation, by ignoring and then not ignoring the newSOA modes in the activation/radiative transfer scheme?

Re: The radiative effects due to SOA calculated in this study are all caused by all SOA (newSOA and internally mixed SOA), not by only newSOA. These effects are the difference in radiation between with and without SOA. The other inputs to

the radiation calculation are all same. The difference in the schemes between with and without organic nucleation is the simulation with the mechanism of new organic particle formation switch on and off. We have described these at the end of Method section.

Fig. S2 caption: “cited from”->“reproduced from”. The figure is currently a screenshot from Andreae et al; incorporating this data into Fig 1 may make the permissions requirements easier?

Re: We have incorporated the observation data into Fig 1(a).

Fig S4, S8: Please use a logarithmic colour scale.

Re: We have changed the Fig S4 and S8 to a logarithmic color scale. Note: the Fig S4 and S8 in the previous version is Fig. S3 and S7 now because we merged Fig S2 to Fig 1(a).

Fig S5 Please state the units more clearly, and explain them in the caption.

Re: We have added the units in the caption in the Fig S4 (which is Fig S5 in the previous version). The unit of 10^{10} m^{-2} is used for the column number concentration shown in the left column of the Figure S4. The unit of 10^{10} m^{-3} is used for the number concentration in the PBL shown in the right column of the Figure S4.

Reviewer #2 (Remarks to the Author):

Zhu et al., Organic aerosol nucleation, climate and land use change: Decrease in radiative forcing

Main claims/findings of the paper are that organic nucleation is an important source of aerosols in the pristine continental areas and during the preindustrial period, and that the increased emissions of α -pinene and its oxidation products (nucleating and forming SOA) are enhancing due to the climate and land use changes. Together these events: organic nucleation, land use and climate, lead to an overall 15 % reduction in radiative forcing when applied together with present day anthropogenic aerosol in the tropics. This paper also demonstrates reproductions of vertical profile of condensation nuclei over the Amazon forest.

I only have more general comments to this work rather than going into details:

- The subject of the study is very complex and the study seems novel at least considering most parts of the work. I'm pleased to see that the effect of climate and land use is included in this work with particle formation happening above the planetary boundary layer. As organic (pure biogenic) nucleation has only been observed and published a few times (Kirkby et al., Nature = lab study and Bianchi et al., Science = field study), models are not accounting it in yet. Even if I consider the methods and results to be novel, I feel that there are too many misinterpretations and misunderstanding with the recent experimental work done in the field of nucleation studies and HOMs. I also find referencing to previous work inadequate and somewhat misleading, especially considering the referencing of experimental publications. Thus, I cannot advise publication in the current form.

- I found that it was difficult to understand if all the work is concerning the vertically integrated values or upper troposphere/PBL conditions only. This should be clarified throughout the manuscript.

Re: Most of results reported in this study are vertically integrated values which are reported as vertically integrated. The only discussion of the concentration in the upper troposphere is the discussion of Fig 1, which shows a profile, with the peak of the aerosol number concentration in the upper troposphere over the Amazon. There are only a few sentences discussing the number concentration in the PBL shown in the supplementary Table 1. All the number concentrations in the PBL shown in the paper have been emphasized with “in the PBL” such as at L127, L177, L178, L239, L267.

- Applying experimental information: The paper claims that the Amazon is the most important region producing new organic particles and has the highest organic nucleation rate primarily because of the high emissions of α -pinene and the high production of HOMs from α -pinene oxidation. My knowledge is that the Amazon is known for its extremely high concentrations and emissions of isoprene, not α -pinene. Even though both α -pinene and isoprene emissions are estimated to increase due to land use and temperature, isoprene is still clearly dominating the BVOCs in the Amazonas. It should be noted in the manuscript that the large fraction of SOA forming HOMs have very different formation pathways, isoprene is mostly oxidized by OH-radicals to form HOM and monoterpenes produce HOM from ozonolysis (e.g. Ehn et al., 2014), OH-radicals (E.g. Berndt et al., 2016) and NO₃-radicals (E.g. Ayres et al., 2015, Yan et al., 2016). Major changes in the oxidation pathways could cause very different results in nucleation, SOA

formation and load. Lots of HOM related work, production mechanisms, yields of different HOMs from various monoterpenes, sesquiterpenes and isoprene (SOAS campaign, Alabama) has been published since the initial Ehn et al, 2014 paper.

Re: The emission of α -pinene is highest in the Amazon as is the emission of isoprene, although the magnitude of the α -pinene emission is one order of magnitude smaller than isoprene emissions. The annual average emission rate of α -pinene and isoprene are shown in the following figure.

Figure. The annual average emission rate of α -pinene (left) and isoprene (right).

We have looked at all the paper you mentioned above about the chemical mechanism. Although there are many pathways to form HOMs, the issue is whether the formed HOMs lead to nucleation or not. Diacyl peroxide is suspected to be the one kind of HOMs to nucleate based on the evidence published in Ziemann (2002). The oxidation product of pinanediol was observed in the new particle formation experiments (Schobesberger et al., 2013) and field measurements of nucleation events (Kulmala et al., 2013). Pinic acid and pinanediol were shown to form new particles through quantum chemistry calculations (Ortega et al., 2016). These four kinds of HOMs are the only ones that we have found with any evidence that they contribute to nucleation and new particle formation. Pinic acid and diacyl peroxide form from the ozonolysis of α -

pinene, and pinanediol and its oxidation product form from an oxidation product of the reaction of α -pinene and OH radical. We didn't find any evidence for the nucleation of HOMs formed from the oxidation of isoprene. As the result, we updated the chemical mechanism used in our model to include the formation of these additional HOMs species (diacyl peroxide, pinic acid, pinanediol and its oxidation product). Further, we updated the nucleation scheme in the model to calculate organic nucleation based on these four kinds of HOMs. We believe the knowledge about the nucleation of HOMs is still limited. The pathways to form HOMs leading to nucleation have large uncertainty. We stated the uncertainty in the discussion section as "The calculation of SOA and RF is uncertain in part because of unclear chemical formation mechanisms. We currently have limited knowledge of the explicit chemical reactions that lead to the production of extremely low volatility HOMs, although there should be many ways that these form. We only included the HOMs with evidence to contribute to nucleation in the model now, but it's possible that there are other species of HOMs will be identified in the future. The use of different HOMs leading to nucleation should be a large uncertainty to the RF estimation associated with organic nucleation." For the SOAS project you mentioned in the comments, we found the publications from this project mainly talk about the pathway to form organic nitrate, but do not mention organic nucleation. We updated the chemical mechanism to form organic nitrate in our model referring to Fisher et al. (2016) which was based on the SOAS project.

Reviewer #1 (Remarks to the Author):

Review of 'Organic aerosol nucleation, climate, and land use change: Decrease in radiative forcing'

Hamish Gordon, 17/10/2018

The authors find a reduction in radiative forcing of 16% when a mechanism for pure organic particle formation is included in their chemistry-climate model. This remains an important topic which has not been widely studied in models.

The authors have made some useful revisions to their manuscript and addressed my previous comments well with extensive additions to the supplementary materials. The new model evaluation is excellent and sets a high standard for future studies. Section S3.1 with its Figure S16 is particularly welcome in the context of my previous comments, and Section S3.2 is also a very interesting addition.

However, the authors do not yet discuss their valuable new work on the uncertainties in nucleation and growth rates in the main text of the paper (neither sections S3.1 nor S3.2 are referred to in the main text). They discuss only the uncertainties associated with formation mechanism for HOMs. The other key uncertainties in the study should be discussed in the main text: nucleation rate temperature dependence, growth rates, BVOC emissions, the very large uncertainties in simulated land use changes, the effect of uncertain pre-industrial fire or other primary particulate emissions on the condensation sink – all of these could change the results drastically.

Nevertheless, the new manuscript is of high quality with a great deal of original material. In my opinion it will be suitable for publication in Nature Communications as soon as lines 292-310 are rewritten to discuss all important uncertainties, as above, and the other details below are addressed.

Specific comments

1. Line 268 "Thus, the inclusion of organic nucleation leads to a 11% smaller IRF of anthropogenic aerosol, which is much smaller than the 27% reduction indicated in ref. 12:"

Please refer, here or at the end of the paragraph, to the large uncertainty in the temperature dependence of the nucleation rates, and to your new supplementary study. The uncertainty in the temperature dependence is still not mentioned in the main text of the manuscript. Also please include a sentence to acknowledge, in the main text of your paper, that Gordon et al found a smaller

forcing reduction when they used the same temperature dependence as you use (17% instead of 27%, from 0.14Wm⁻² vs 0.22Wm⁻² on page 4 of the paper).

It's also true that the overall anthropogenic aerosol forcing in Gordon et al (2016) is quite a lot less negative than in your study. This might also help explain why the percentage change you find is smaller. In addition, it struck me that the sensitivity of the forcing reduction you calculate to the temperature dependence is a bit lower than in Gordon (2016). I thought this was interesting, and you might also want to comment.

2. Line 182: 'larger in a factor of 11.4 and 7.6 than PIall'

The very large increase in the number concentration of newSOA in the accumulation mode surprised me a little. However, the very useful evaluation of OC mass in the present day suggests SOA is underestimated by quite a bit. As it is also likely to be underestimated in the PI atmosphere, in reality one would expect a smaller increase in the accumulation mode concentration of newSOA in the PD compared to the PI. Please comment briefly that the changes in number concentration of accumulation-mode newSOA may be sensitive to the absolute amount of newSOA and refer to the supplementary materials. Also suggest rephrase sentence to "larger than PIall by factors of 11.4 and 7.6 respectively."

3. Line 295: "we included those HOMs that have been shown to contribute to nucleation."

Please rephrase to "we included the HOMs that have been shown to contribute to nucleation with quantum chemical calculations" so that the reader knows that HOMs formed by autoxidation are not included.

4. Supplement Line 197: "organic nucleation always occurs within the PBL in the Gordon et al. study"

This is not technically correct, organic nucleation happens mostly near the surface as it is temperature independent, but it is not restricted to the boundary layer (in either Gordon et al 2016 or Gordon et al 2017).

Reviewer #2 (Remarks to the Author):

I thank the authors for the informative response to both of the reviews. All my questions and comments have been satisfactorily addressed in your point to point reply. I have no further comments concerning this manuscript at this point.

Response to the reviewers' comments on “Organic aerosol nucleation, climate, and land use change: Decrease in radiative forcing”

Jialei Zhu¹, Joyce E. Penner^{1*}, Fangqun Yu², Sanford Sillman¹, Meinrat O. Andreae^{3,4}, Hugh Coe⁵

1 Department of Climate and Space Sciences and Engineering, University of Michigan, Ann Arbor, Michigan 48109, USA

2 Atmospheric Sciences Research Center, State University of New York at Albany, Albany, New York 12203, USA

3 Biogeochemistry Department, Max Planck Institute for Chemistry, Mainz, Germany

4 Scripps Institution of Oceanography, University of California San Diego, La Jolla, CA 92093, USA

5 School of Earth and Environmental Sciences, University of Manchester, Manchester M13 9PL, UK

Corresponding Author:

Joyce E. Penner

Address: Space Research Building, University of Michigan, 2455 Hayward Street, Ann Arbor, Michigan 48109, USA

Telephone Number: (734) 936-0519

Email: penner@umich.edu

We thank editor and two reviewers very much for the important comments. Replies to these comments are listed point by point as below. We also highlight the related change in the main text and supplementary.

Reviewers' comments:

Reviewer #1 (Remarks to the Author):

Review of 'Organic aerosol nucleation, climate, and land use change: Decrease in radiative forcing'

Hamish Gordon, 17/10/2018

The authors find a reduction in radiative forcing of 16% when a mechanism for pure organic particle formation is included in their chemistry-climate model. This remains an important topic which has not been widely studied in models.

The authors have made some useful revisions to their manuscript and addressed my previous comments well with extensive additions to the supplementary materials. The new model evaluation is excellent and sets a high standard for future studies. Section S3.1 with its Figure S16 is particularly welcome in the context of my previous comments, and Section S3.2 is also a very interesting addition.

However, the authors do not yet discuss their valuable new work on the uncertainties in nucleation and growth rates in the main text of the paper (neither sections S3.1 nor S3.2 are referred to in the main text). They discuss only the uncertainties associated with formation mechanism for HOMs. The other key uncertainties in the study should be discussed in the main text: nucleation rate temperature dependence, growth rates, BVOC emissions, the very large uncertainties in simulated land use changes, the effect of uncertain pre-industrial fire or other primary particulate emissions on the condensation sink – all of these could change the results drastically.

Nevertheless, the new manuscript is of high quality with a great deal of original material. In my opinion it will be suitable for publication in Nature Communications as soon as lines 292-310 are rewritten to discuss all important uncertainties, as above, and the other details below are addressed.

RE: Thanks for your approval of our reply and revision in the last version. According to your comments, we have added some discussion of the uncertainty associated with the temperature dependence of the nucleation rate (based on S3.1), the growth rate of newSOA (based on S3.2), fire emission in pre-industrial period and some other uncertainties such as land use change in the end of the “Implications and Discussion” section (after line 314).

Specific comments

1. Line 268 “Thus, the inclusion of organic nucleation leads to a 11% smaller IRF of anthropogenic aerosol, which is much smaller than the 27% reduction indicated in ref. 12:”

Please refer, here or at the end of the paragraph, to the large uncertainty in the temperature dependence of the nucleation rates, and to your new supplementary study. The uncertainty in the temperature dependence is still not mentioned in the main text of the manuscript. Also please include a sentence to acknowledge, in the main text of your paper, that Gordon et al found a smaller forcing reduction when they used the same temperature dependence as you use (17% instead of 27%, from 0.14Wm^{-2} vs 0.22Wm^{-2} on page 4 of the paper).

It’s also true that the overall anthropogenic aerosol forcing in Gordon et al (2016) is quite a lot less negative than in your study. This might also help explain why the percentage change you find is smaller. In addition, it struck me that the sensitivity of the forcing reduction you calculate to the temperature dependence is a bit lower than in Gordon (2016). I thought this was interesting, and you might also want to comment.

RE: We have added a mention the 17% reduction when Gordon et al. (2016) used the same temperature dependence as we have and also refer to the supplementary S3.1 (Line 270-273)

2. Line 182: ‘larger in a factor of 11.4 and 7.6 than PI_{all} ’

The very large increase in the number concentration of newSOA in the accumulation mode surprised me a little. However, the very useful evaluation of OC mass in the present day suggests SOA is underestimated by quite a bit. As it is also likely to be underestimated in the PI atmosphere, in reality one would expect a smaller increase in the accumulation mode concentration of newSOA in the PD compared to the PI. Please comment briefly that the changes in number concentration of accumulation-mode newSOA may be sensitive to the absolute amount of newSOA and refer to the supplementary materials. Also suggest rephrase sentence to “larger than PIall by factors of 11.4 and 7.6 respectively.”

RE: We have added a sentence of “The large increase in the number concentration of accumulation mode newSOA in the PD may be caused by the underestimate of SOA in our model (Supplementary S1), which leads to a very low baseline in the number of newSOA in the accumulation mode in PIall.” as a comment to this large increase in Lines 182-185. Also we have changed the sentence to “larger than that in PIall by factors of 11.4 and 7.6, respectively.” as you suggested.

3. Line 295: “we included those HOMs that have been shown to contribute to nucleation.”

Please rephrase to “we included the HOMs that have been shown to contribute to nucleation with quantum chemical calculations” so that the reader knows that HOMs formed by autoxidation are not included.

RE: We have changed this sentence to “we included those HOMs that have been shown to contribute to nucleation with quantum chemical calculations”.

4. Supplement Line 197: “organic nucleation always occurs within the PBL in the Gordon et al. study”

This is not technically correct, organic nucleation happens mostly near the surface as it is temperature independent, but it is not restricted to the boundary layer (in either Gordon et al 2016 or Gordon et al 2017).

RE: We have changed this sentence to “organic nucleation mainly occurs near the surface when it is independent of temperature in the Gordon et al. study”. (Line 179-182 in the supplementary)

Reviewer #2 (Remarks to the Author):

I thank the authors for the informative response to both of the reviews. All my questions and comments have been satisfactorily addressed in your point to point reply. I have no further comments concerning this manuscript at this point.

RE: Thanks for your approval of our reply and revision in the last version.